# SILVER STEPSIZE FOR FASTER ZEROTH-ORDER OPTIMIZATION

## ABSTRACT

We study gradient-free minimization of smooth convex functions via *Silver stepsizes*, a non-monotone 2-adic schedule that accelerates gradient descent, composed with two-point zeroth-order (ZO) estimators on a smoothed objective. We show that the *multi-step Lyapunov (Silver) analysis* carries over when exact gradients are replaced by *conditionally unbiased* two-point estimators, with a stochastic tax that reduces to a *quadratic variance* term. We control this term under a fixed query budget by an *orthogonal-on-spikes* batching policy $B_t \propto \alpha_t$, which is *budget-optimal*. Empirically, we validate our approach on numerical quadratics across different conditioning regimes and *MeZO*-style forward-only fine-tuning of RoBERTa-large on GLUE tasks (SST-2, RTE), ZO-SILVER reduces evaluation loss faster than tuned constant-LR MeZO under the same query budget.

## 1 INTRODUCTION

Zeroth-order (ZO, derivative-free) optimization addresses the common setting where we can query function values but cannot reliably obtain gradients: the model is a black box, gradients are prohibitively expensive or noisy, or we wish to optimize through a non-differentiable system (e.g., simulators, private APIs). This regime occurs across machine learning and scientific computing: hyperparameter and architecture tuning, black-box adversarial attacks, policy search and evolution strategies in RL, and large-model fine-tuning under tight memory budgets (Larson et al., 2019; Flaxman et al., 2005; Duchi et al., 2015; Shamir, 2017; Salimans et al., 2017; Malladi et al., 2023a).

**Families of ZO estimators.** Modern ZO methods approximate gradients from function values using structured perturbations. (i) *One-point bandit smoothing* forms an unbiased estimator of the gradient of a smoothed objective from a single evaluation (Flaxman et al., 2005). (ii) *Two-point estimators*—our focus—use symmetric differences $f(x + \mu u) - f(x - \mu u)$ along a random direction $u$, achieving strictly better variance/rates and minimax-optimal guarantees for smooth convex objectives (Duchi et al., 2015; Shamir, 2017; Nesterov & Spokoiny, 2017). (iii) *Coordinate-wise finite differences* estimate partial derivatives one coordinate at a time (often $2d$ queries per gradient) and are widely used in black-box deep learning (e.g., ZOO attacks) (Chen et al., 2017). (iv) *SPSA* perturbs all coordinates simultaneously using Rademacher noise and recovers a two-evaluation gradient proxy with strong SA-style guarantees (Spall, 1992). (v) *Orthogonal batches* sample $B$ mutually orthonormal directions (Stiefel manifold) per iteration; this reduces the estimator variance at fixed budget and unifies several schemes, including spherical smoothing and coordinate descent (Kozak et al., 2023; Feng & Wang, 2023).

**Core bottlenecks in ZO.** ZO estimators introduce a bias–variance tradeoff via the smoothing radius $\mu$ and sampling distribution. Even for smooth convex objectives, the best-known two-point schemes incur a statistical floor that scales with dimension under noisy queries; controlling the *variance accumulation* across iterations is the central algorithmic challenge (Duchi et al., 2015; Shamir, 2017; 2013; Jamieson et al., 2012).

**An acceleration lever: stepsize hedging (Silver).** Independently of estimator design, recent work shows that carefully structured *stepsizes* alone can accelerate plain gradient descent on smooth convex functions. The *Silver stepsize schedule* is a simple, explicit, fractal sequence with a 2-adic block structure. It admits a *multi-step Lyapunov certificate* ("Silver identity") which gives a convergence

rate of $O\big(\varepsilon^{-\log_\rho 2}\big) = O\big(\varepsilon^{-0.7864}\big)$ iterations for gradient descent, where $\rho = 1 + \sqrt{2}$ is the silver ratio (Altschuler & Parrilo, 2023a;b; 2024). Intuitively, the schedule interleaves small steps with periodic "spikes" whose algebraic cancellation accelerates net progress across blocks.

**This work: composing Silver with two-point ZO on smoothed objectives.** We bring these strands together. We run the Silver schedule on a *smoothed* objective $h = f_\mu/L$ (blockwise-constant $\mu$), and replace exact gradients by unbiased *symmetric two-point* estimators for $\nabla f_\mu$ along *orthonormal* batches of directions. The Silver identity's linear noise terms cancel in expectation, so the entire stochastic tax collapses to an explicit *quadratic variance term*, which we control by aligning batch size with the stepsize spikes ($B_t \propto \alpha_t$, capped at $d$). This *orthogonal-on-spikes* policy concentrates averaging where it matters most while keeping the total query budget fixed.

**Motivation** Two-point estimators are unbiased for $\nabla f_\mu$ (not $\nabla f$), making the smoothed problem $f_\mu$ the right analytical object. The Silver identity is robust to *conditionally unbiased* inexact gradients and only pays the quadratic term from the terminal square in the certificate—precisely what batching and blockwise $\mu$ can control. Orthogonal directions improve constants without complicating the analysis or the memory footprint (Kozak et al., 2023; Feng & Wang, 2023).

We make the following contributions in this work.

- Silver-on-smoothing with two-point ZO: We adapt the Silver multi-step analysis to $h = f_\mu/L$ with symmetric two-point estimators, showing the identity carries overwith a single *variance aggregation* term $\sum_t \alpha_t^2 \mathbb{E}\|\zeta_t\|^2$ (no linear noise term).
- Variance control via orthogonal-on-spikes batching: Under a fixed query budget per block, we prove that allocating batch sizes proportional to the Silver steps ($B_t \propto \alpha_t$, capped at $d$) optimally controls $\sum_t \alpha_t^2/B_t$ (Cauchy–Schwarz tightness), and we instantiate this with Stiefel sampling.
- High-probability bounds via Freedman: We give a simple high-probability translation of the Silver identity with martingale differences, giving dimension-aware tails in terms of the predictable quadratic variation.
- Practical ZO for LLM fine-tuning: We apply the method to MeZO-style forward-only full-parameter fine-tuning and discuss practical details (direction orthogonalization, clipping, memory footprint.) (Malladi et al., 2023a; Hu et al., 2021; Dettmers et al., 2023).

**Organization.** Section 3.1 states the formal setup and notation; Section 3.2 summarizes the Silver schedule and the specific properties we use. Section 4 develops the inexact-gradient Silver identity for two-point ZO on $f_\mu$ and the variance control via orthogonal-on-spikes batching. Experiments appear in Section 5.

## 2 RELATED WORK

**Derivative-free / zeroth-order optimization.** Classical DFO covers direct-search, model-based trust-region, and interpolation methods; recent surveys unify these with randomized finite-difference estimators used in ML (Larson et al., 2019). For convex ZO with random directions, one-point bandit smoothing dates to Flaxman et al. (2005). Two-point estimators achieve optimal rates in smooth/stochastic and adversarial settings (Duchi et al., 2015; Shamir, 2017). Nesterov & Spokoiny (2017) give a self-contained analysis with explicit smoothing constants. Building on this line of work, MeZO (Malladi et al., 2023a) brings two-point, forward-only ZO into LLM fine-tuning, showing that competitive adaptation is possible with inference-level memory (no backprop activations). In this work, we analyze with uniform sphere sampling for slightly tighter dimension-dependent estimation variance at high dimension.

**Estimator families and variance reduction.** Coordinate-wise finite differences (up to $2d$ queries/gradient) are common in black-box deep learning, e.g., ZOO (Chen et al., 2017). SPSA provides a two-evaluation coordinate-free estimator rooted in stochastic approximation (Spall, 1992). Sampling *orthogonal* directions (Stiefel manifold) reduces variance and unifies spherical and coordinate schemes (Kozak et al., 2023); refined bounds appear in Feng & Wang (2023). Variance-reduced ZO methods (e.g., ZO-SVRG/SPIDER-SZO) are complementary and can be combined with our blockwise policy (Ji et al., 2019; Fang et al., 2018).

**Zeroth-order smoothing and two-point estimators.** Ball/sphere and Gaussian smoothing with two-point estimators are classical; see Flaxman et al. (2005) (one-point bandit smoothing), Duchi et al. (2015); Shamir (2017) (two-point optimal rates), and Nesterov & Spokoiny (2017) (Gaussian smoothing with explicit moment and bias constants). We emphasize the uniform *ball/sphere* pair, which gives dimension-friendly bias constants and a clean gradient identity.

**Stepsize hedging / Silver schedule.** The Silver schedule is a simple explicit fractal stepsize sequence that accelerates plain gradient descent in both strongly convex and smooth convex regimes. The analysis hinges on a multi-step descent identity and 2-adic structure; see Altschuler & Parrilo (2023a;b; 2024) for the arXiv and final journal versions. The rate $T^{-\log_\rho 2}$ with $\rho = 1 + \sqrt{2}$ lies between classical $O(\varepsilon^{-1})$ and Nesterov's $O(\varepsilon^{-1/2})$.

# 3 PRELIMINARIES

## 3.1 PROBLEM SETUP AND NOTATION

We minimize a convex $L$-smooth function $f : \mathbb{R}^d \to \mathbb{R}$ with minimizer $x^\star$. We adopt the standard uniform-ball smoothing

$$f_\mu(x) := \mathbb{E}_{v \sim \text{Unif}(\mathbb{B}^d)} f(x + \mu v), \qquad h(x) := f_\mu(x)/L,$$

so that $h$ is 1-smooth and convex. We use the symmetric two-point estimator because it enjoys sharper variance/rate guarantees in smooth convex problems (Duchi et al., 2015; Shamir, 2017; Nesterov & Spokoiny, 2017).

**Uniform-ball smoothing and the sphere gradient identity.** Let $v \sim \text{Unif}(\mathbb{B}^d)$ and $u \sim \text{Unif}(\mathbb{S}^{d-1})$, and define $f_\mu(x) := \mathbb{E}_v f(x + \mu v)$. Then $f_\mu$ is convex and $L$-smooth and

$$\nabla f_\mu(x) = \frac{d}{\mu} \mathbb{E}_u\big[f(x + \mu u) u\big]. \tag{1}$$

And,

$$|f_\mu(x) - f(x)| \leq \tfrac{L}{2}\mu^2 \mathbb{E}\|v\|^2 = \tfrac{L}{2}\mu^2 \cdot \tfrac{d}{d+2}.$$

Moreover (proofs in the appendix),

$$\|\nabla f_\mu(x) - \nabla f(x)\| \leq \tfrac{L}{2} d \mu. \tag{2}$$

For comparison, under *Gaussian* smoothing, $\|\nabla f_\mu(x) - \nabla f(x)\| \leq \frac{L}{2}(d+3)^{3/2}\mu$ (Nesterov & Spokoiny, 2017, Lemma 3).

*Remark* 3.1 (Default smoothing and unbiasedness). Throughout we define $f_\mu(x) = \mathbb{E}_{v \sim \text{Unif}(\mathbb{B}^d)} f(x + \mu v)$. For this choice,

$$\nabla f_\mu(x) = \frac{d}{\mu} \mathbb{E}_{u \sim \text{Unif}(\mathbb{S}^{d-1})}[f(x + \mu u) u],$$

so both the one-point $\frac{d}{\mu}f(x + \mu u)u$ and the symmetric two-point $\frac{d}{2\mu}(f(x + \mu u) - f(x - \mu u))u$ estimators are *unbiased* for $\nabla f_\mu(x)$. This identity goes back to the divergence-theorem proof used in bandit smoothing (e.g., Flaxman et al. (2005)).[1]

**Iteration, stepsizes, and batching.** We run a Silver block of length $N = 2^k - 1$ with stepsizes $\{\alpha_t\}_{t=0}^{N-1}$ (Section 3.2), update

$$x_{t+1} = x_t - \frac{\alpha_t}{L}\widehat{g}_t, \qquad \widehat{g}_t = \frac{d}{2\mu B_t}\sum_{i=1}^{B_t}\big(f(x_t + \mu v_{t,i}) - f(x_t - \mu v_{t,i})\big)v_{t,i},$$

and use *orthogonal-on-spikes* batching $B_t = \min\{d, \lceil c_B \alpha_t \rceil\}$ with $V_t = [v_{t,1}, \dots, v_{t,B_t}] \in \text{St}(d, B_t)$ drawn via thin QR of a Gaussian matrix (Haar on the Stiefel manifold). Each step costs $2B_t$ function queries. Unless stated otherwise, we assume access to exact function values or conditionally zero-mean value noise so that $\mathbb{E}[\widehat{g}_t \mid \mathcal{F}_{t-1}] = \nabla f_\mu(x_t)$ with $\mathcal{F}_{t-1}$ the natural filtration up to the start of iteration $t$ (so $x_t, \alpha_t$ are $\mathcal{F}_{t-1}$-measurable).

---

[1]We work with the *ball* definition of $f_\mu$ for tighter bias; we only use the *sphere* for the estimator.

## 3.2 Silver stepsizes primer

Let $\rho := 1 + \sqrt{2}$ and let $v(i)$ be the 2-adic valuation of $i \in \mathbb{N}$. We use the explicit schedule

$$\alpha_i = 1 + \rho^{v(i)-1}, \qquad i = 1, 2, \ldots$$

(optionally scaled and/or clipped in practice). This closed form matches the recursive construction and gives a fractal 2-adic block structure.(Altschuler & Parrilo, 2023a;b; 2024)

For a block of length $N = 2^k - 1$ and a 1-smooth convex objective $h$, Altschuler & Parrilo (2023b) establish a multi-step Lyapunov identity which implies

$$h(x_N) - h^\star \leq r_k \|x_0 - x^\star\|^2,$$

with explicit

$$r_k := \frac{1}{1 + \sqrt{4\rho^{2k} - 3}} \leq \frac{1}{2\rho^{\log_2 n}} = \frac{1}{2n^{\log_2 \rho}} = O(n^{-1.2716})$$

for $\rho = 1 + \sqrt{2}$ (the silver ratio). This rate improves upon the classical Gradient Descent convergence of $O(1/n)$, positioning itself as an intermediary between this baseline and the accelerated convergence rate $O(1/n^2)$ of Nesterov's method, which is known to be optimal for first-order smooth convex optimization (Nesterov, 1983; Nemirovsky & Yudin, 1983). This improvement is achieved without modifying the algorithmic structure or introducing momentum terms, only through an appropriate choice of step sizes. Consequently, after $N = \Theta(2^k)$ steps, gradient descent with Silver stepsizes reaches error $\varepsilon$ in $O(\varepsilon^{-\log_\rho 2}) = O(\varepsilon^{-0.7864})$ iterations, strictly improving upon the classical $O(1/\varepsilon)$ rate for smooth convex objectives (Altschuler & Parrilo, 2023a;b; 2024). In our analysis we apply this identity to $h = f_\mu/L$ and rely only on:

1. the block guarantee $h(x_N) - h^\star \leq r_k \|x_0 - x^\star\|^2$;
2. the sum-of-steps property $\sum_{t=0}^{N-1} \alpha_t = \Theta(\rho^k)$;
3. robustness to *conditionally unbiased* inexact gradients, which adds exactly $\sum_t \alpha_t^2 \mathbb{E}\|\zeta_t\|^2$ to the RHS (no linear noise term).

# 4 ZO-SILVER: Algorithm And Theoretical Analysis

## Roadmap of this section

We state the blockwise guarantees first: (i) an expectation-level *one-block* bound under two-point ZO on the smoothed objective with Silver steps; (ii) a *budget-aligned* specialization under orthogonal-on-spikes batching; (iii) a *multi-block* (restart) bound; and (iv) a *high-probability* version via Freedman. We then present the algorithm (with Stiefel sampling) and the minimal ingredients (unbiasedness, second moment, inexact-Silver identity, and the variance-optimal batching proposition).

## 4.1 Assumptions and oracle model

**Problem class and oracle** $f : \mathbb{R}^d \to \mathbb{R}$ is convex and $L$-smooth. We query a value oracle that returns either exact $f(x)$ or $f(x) + \xi$ with conditionally zero-mean noise ($\mathbb{E}[\xi \mid x] = 0$) and finite variance. We adopt uniform-ball smoothing $f_\mu(x) = \mathbb{E}_{v \sim \text{Unif}(\mathbb{B}^d)} f(x + \mu v)$ and define $h := f_\mu/L$, so that $h$ is 1-smooth and convex. Within each Silver block, $\mu$ is fixed. At iteration $t$, we form the symmetric two-point estimator with $B_t$ unit directions $V_t = [v_{t,1}, \ldots, v_{t,B_t}] \in \text{St}(d, B_t)$ sampled independently of the past and use the same batch for $\pm\mu$ queries.

## 4.2 Ingredients (unbiasedness, second moment, inexact Silver, batching)

**Lemma 4.1** (Second moment: uniform sphere, symmetric two-point). *Let $f \in C_L^{1,1}$, $u \sim \text{Unif}(\mathbb{S}^{d-1})$, and $\widehat{g}(x; \mu, u) = \frac{d}{2\mu}(f(x + \mu u) - f(x - \mu u))u$. Then*

$$\mathbb{E}\|\widehat{g}(x; \mu, u) - \nabla f_\mu(x)\|^2 \leq C_{\text{sig}} d \|\nabla f(x)\|^2 + C_{\text{curv}} d^2 L^2 \mu^2,$$

*with $(C_{\text{sig}}, C_{\text{curv}}) = (2, \frac{1}{2})$. Averaging any $B \geq 1$ unit directions gives a $1/B$ reduction. Using $B$ orthonormal directions (Stiefel sampling) preserves the $1/B$ factor and improves constants in practice (Kozak et al., 2023; Feng & Wang, 2023).*

These constants are tight up to lower-order terms for two-point ZO under $L$-smoothness; see the elementary proof in the appendix and the companion derivation we follow. The proof is in the Appendix A.

**Filtration and conditional unbiasedness.** Let $\mathcal{F}_{t-1}$ denote the $\sigma$-field generated by all randomness up to the *start* of iteration $t$ (so $x_t$ and $\alpha_t$ are $\mathcal{F}_{t-1}$-measurable). At iteration $t$, sample fresh directions $V_t$ independently of $\mathcal{F}_{t-1}$ (uniform on $\mathbb{S}^{d-1}$ or Haar on $\mathrm{St}(d, B_t)$), and evaluate $f$ exactly (or with conditionally zero-mean noise) using the same batch for the $\pm\mu$ queries. For the symmetric two-point estimator we then have

$$\mathbb{E}[\widehat{g}_t \mid \mathcal{F}_{t-1}] = \nabla f_\mu(x_t), \qquad \zeta_t := \frac{1}{L}\big(\widehat{g}_t - \nabla f_\mu(x_t)\big), \qquad \mathbb{E}[\zeta_t \mid \mathcal{F}_{t-1}] = 0.$$

With predictable stepsizes $(\alpha_t)$, define the predictable quadratic variation

$$V := \sum_{t=0}^{N-1} \alpha_t^2 \, \mathbb{E}[\zeta_t \zeta_t^\top \mid \mathcal{F}_{t-1}].$$

**Lemma 4.2** (Martingale square identity). *Let $\{\zeta_t\}_{t=0}^{N-1}$ be a square-integrable vector MDS adapted to $(\mathcal{F}_t)$, so $\mathbb{E}[\zeta_t \mid \mathcal{F}_{t-1}] = 0$, and let $\{\alpha_t\}$ be deterministic (or merely $\mathcal{F}_{t-1}$-measurable). Then*

$$\mathbb{E}\Big\| \sum_{t=0}^{N-1} \alpha_t \, \zeta_t \Big\|^2 = \sum_{t=0}^{N-1} \alpha_t^2 \, \mathbb{E}\|\zeta_t\|^2.$$

*Proof.* Expand the square; for $s < t$, $\mathbb{E}\langle \zeta_s, \zeta_t\rangle = \mathbb{E}\big[\langle \zeta_s, \mathbb{E}[\zeta_t \mid \mathcal{F}_{t-1}]\rangle\big] = 0$, since $\zeta_s$ is $\mathcal{F}_{t-1}$-measurable. $\square$

### 4.3 MAIN RESULTS

**Lemma 4.3** (Inexact Silver, expectation level). *Let $h = f_\mu/L$ (so $h$ is 1-smooth and convex) and suppose $x_{t+1} = x_t - \alpha_t(\nabla h(x_t) + \zeta_t)$ with $\mathbb{E}[\zeta_t \mid \mathcal{F}_{t-1}] = 0$. For a Silver block $N = 2^k - 1$,*

$$\mathbb{E}\big[h(x_N) - h^\star\big] \leq r_k \, \mathbb{E}\|x_0 - x^\star\|^2 + \sum_{t=0}^{N-1} \alpha_t^2 \, \mathbb{E}\|\zeta_t\|^2.$$

*Proof.* See Appendix. $\square$

**Variance-optimal batching under a query budget.** We motivate the batching policy with the following variance-related observation.

**Proposition 4.4** (Optimal allocation of directions under a query budget). *Fix nonnegative weights $\{\alpha_t\}_{t=0}^{N-1}$ and a budget $Q > 0$ of function queries per block. With symmetric two-point queries, $Q = 2\sum_t B_t$. Then, for any $B_t > 0$,*

$$\sum_{t=0}^{N-1} \frac{\alpha_t^2}{B_t} \geq \frac{\big(\sum_{t=0}^{N-1} \alpha_t\big)^2}{\sum_{t=0}^{N-1} B_t} = \frac{2\big(\sum_t \alpha_t\big)^2}{Q},$$

*with equality iff $B_t \propto \alpha_t$. In particular, the policy $B_t = \min\{d, \lceil c_B \alpha_t\rceil\}$ is (up to the cap and integrality) optimal for a given budget.*

*Proof.* By Cauchy–Schwarz, $\big(\sum \frac{\alpha_t^2}{B_t}\big)\big(\sum B_t\big) \geq \big(\sum \alpha_t\big)^2$. Substitute $\sum B_t = Q/2$. $\square$

**Theorem 4.5** (One block, expectation). *Assume $f : \mathbb{R}^d \to \mathbb{R}$ is convex and $L$-smooth, and fix a Silver block of length $N = 2^k - 1$ with steps $\{\alpha_t\}_{t=0}^{N-1}$. Let $f_\mu$ be the uniform-ball smoothing, $h = f_\mu/L$, and define the symmetric two-point estimator averaged over $B_t$ unit directions (orthonormal columns $V_t \in \mathrm{St}(d, B_t)$ drawn independently of $\mathcal{F}_{t-1}$)*

$$\widehat{g}_t = \frac{d}{2\mu B_t} \sum_{i=1}^{B_t} \Big(f(x_t + \mu v_{t,i}) - f(x_t - \mu v_{t,i})\Big) v_{t,i}, \qquad x_{t+1} = x_t - \frac{\alpha_t}{L} \widehat{g}_t.$$

*Then*

$$\mathbb{E}\big[f(x_N) - f^\star\big] \le r_k\,L\,\mathbb{E}\|x_0 - x^\star\|^2 + \sum_{t=0}^{N-1} \frac{\alpha_t^2}{B_t}\left(\frac{1}{2}d^2 L\,\mu^2 + \frac{2d}{L}\,\mathbb{E}\|\nabla f(x_t)\|^2\right) \;+\; \frac{L}{2}\mu^2\frac{d}{d+2}.$$

*In particular, if $\|x_t - x^\star\| \le R$, then $\|\nabla f(x_t)\| \le LR$ and*

$$\mathbb{E}\big[f(x_N) - f^\star\big] \;\le\; r_k\,L\,R^2 \;+\; \left(2d\,R^2 + \tfrac{1}{2}d^2\mu^2\right)\sum_{t=0}^{N-1}\frac{\alpha_t^2}{B_t} \;+\; L\,\mu^2\,\frac{d}{d+2}.$$

*Proof sketch.* We apply the inexact-gradient Silver identity to $h = f_\mu/L$ with $\zeta_t = (\widehat{g}_t - \nabla f_\mu(x_t))/L$ (conditionally unbiased, so the identity has no linear noise term). Use the two-point second-moment bound plus averaging-by-$B_t$, then convert from $h$ to $f$ using the value-bias of $f_\mu$. (see Appendix.)  □

**Budget-aligned specialization.** Let $B_t = \min\{d, \lceil c_B\,\alpha_t\rceil\}$ (orthogonal-on-spikes), and write $\alpha_{\max} = \max_t \alpha_t$. Then

**Proposition 4.6** (Variance aggregation under $B_t \propto \alpha_t$).

$$\sum_{t=0}^{N-1}\frac{\alpha_t^2}{B_t} \;\le\; \frac{1}{c_B}\sum_{t=0}^{N-1}\alpha_t \;+\; \frac{\alpha_{\max}}{d}\sum_{t:\,\alpha_t > d/c_B}\alpha_t.$$

*In particular, if $d \ge c_B\alpha_{\max}$ (cap inactive), then $\sum_t \alpha_t^2/B_t = (1/c_B)\sum_t \alpha_t = \Theta(\rho^k/c_B)$.*

**Corollary 4.7** (Per-block calibration of $\mu$ and $c_B$). *If $\frac{\rho^k}{c_B}\cdot\left(\frac{1}{2}d^2 L\mu^2 + 2dLR^2\right) \le \varepsilon\,r_k\,LR^2$, then*

$$\mathbb{E}[f(x_N) - f^\star] \;\le\; (1+\varepsilon)\,r_k\,LR^2 \;+\; \frac{L}{2}\mu^2\frac{d}{d+2}.$$

*A sufficient choice is $c_B \ge \frac{2d\,\rho^k}{\varepsilon r_k}$ and $\mu^2 \le \frac{2\varepsilon r_k}{d^2}\frac{R^2}{\rho^k}$.*

**Theorem 4.8** (Multi-block restarts). *Run blocks $j = 1,\dots,J$ with lengths $N_j = 2^{k_j} - 1$ and radii $\mu_j$ (each fixed within the block), using $B_t = \min\{d, \lceil c_B\alpha_t\rceil\}$. If $\|x_t - x^\star\| \le R$ across the run, then*

$$\mathbb{E}\big[f(x_{T_J}) - f^\star\big] \;\le\; LR^2\sum_{j=1}^{J} r_{k_j} \;+\; \frac{1}{c_B}\sum_{j=1}^{J}\left(\tfrac{1}{2}d^2 L\mu_j^2 + 2d\,LR^2\right)\rho^{k_j} \;+\; \frac{L}{2}\mu_J^2\frac{d}{d+2}.$$

*Remark* 4.9 (Empirical status of Silver in first-order GD). As far as we are aware, the original Silver papers and their support material emphasize theoretical certificates, and do not provide systematic first-order empirical benchmarks. Discussions on empirical observations and generalizations (e.g., proximal/projected GD) are included, but a standardized FO benchmark suite on Silver vs. standard schedules has not yet emerged. See Altschuler & Parrilo (2023a;b; 2024); Parrilo (2024); Altschuler & Parrilo (2023c); Bok & Altschuler (2024).

## 4.4 ALGORITHM

---

**Algorithm 1** ZO-SILVER: block-constant smoothing + orthogonal-on-spikes batching

---

1: **Input:** block length $N = 2^k - 1$, radius $\mu > 0$, Silver steps $\{\alpha_t\}$, cap $d$, batching constant $c_B > 0$
2: **for** $t = 0,\dots,N-1$ **do**
3:    Stepsize $\eta_t = \alpha_t/L$;   Batch $B_t = \min\{d, \lceil c_B\,\alpha_t\rceil\}$
4:    Sample $V_t = [v_{t,1},\dots,v_{t,B_t}] \in \mathrm{St}(d, B_t)$ (orthonormal columns; e.g., thin QR of a Gaussian matrix)
5:    $\widehat{g}_t = \dfrac{d}{2\mu B_t}\sum_{i=1}^{B_t}\big(f(x_t + \mu v_{t,i}) - f(x_t - \mu v_{t,i})\big)v_{t,i}$
6:    $x_{t+1} = x_t - \eta_t\,\widehat{g}_t$

---

**Sampling orthonormal directions.** A simple implementation samples $G \in \mathbb{R}^{d \times B_t}$ with i.i.d. $\mathcal{N}(0,1)$ entries and sets $V_t$ to the $Q$ factor of the thin QR decomposition $G = V_t R$, giving $V_t \in \mathrm{St}(d, B_t)$ with Haar-distributed columns.

**Corollary 4.10** (Per-block calibration of $\mu$ and batching). *Fix a block of length $N = 2^k - 1$ with $B_t = \min\{d, \lceil c_B \alpha_t \rceil\}$. Assume $\|\nabla f(x_t)\| \le LR_t$ along the block (e.g., by projection or local boundedness). If we choose $\mu$ and $c_B$ to satisfy $\frac{\rho^k}{c_B} \cdot \left( \frac{1}{2} d^2 L \mu^2 + 2dLR_t^2 \right) \le \varepsilon \cdot r_k LR_t^2$, then the block guarantee becomes*

$$\mathbb{E}[f(x_N) - f^\star] \le (1 + \varepsilon) \, r_k LR_t^2 + \tfrac{L}{2} \mu^2 \frac{d}{d+2}.$$

*Equivalently, one sufficient choice is $c_B \ge \frac{2d \, \rho^k}{\varepsilon \, r_k}$ and $\mu^2 \le \frac{2\varepsilon \, r_k}{d^2} \cdot \frac{R_t^2}{\rho^k}$.*

*Remark* 4.11. In addition to the expectation-level bounds presented, we present high-probability results (see Appendix), to further justify our design choices, in particular the batching policy adopted.

# 5 EXPERIMENTS

This section evaluates on two settings: (i) controlled, strongly-convex *quadratics* varying condition number and dimension, and (ii) *forward-only* MeZO-style fine-tuning on GLUE tasks (SST-2, RTE) using RoBERTa-large. Across both, we enforce budget-fairness (same number of function evaluations).

**Two-point ZO and budget matching.** Each iteration uses the symmetric two-point estimator (Sec. 3.1); querying $B_t$ directions costs $2B_t$ function calls. We report and *match* $Q = 2\sum_t B_t$ across methods. We keep smoothing $\mu$ *block-constant* and use the same $\mu$ across baselines to isolate scheduling effects.

We compare against a tuned **ZO-GD (constant LR)** baseline that uses the *same* two-point estimator, smoothing $\mu$, and matched query budget.

We first start by a quadratic test case, across different conditionings and dimensions, we then evaluate on two GLUE classification tasks: *SST-2* (binary sentiment) and *RTE* (recognizing textual entailment). We adopt the MeZO forward-only two-point estimator and compare (i) constant learning rate baselines to (ii) ZO-SILVER with clipped Silver stepsizes. We match the per-step query budget across both methods (Silver uses the same base LR, and the constant-LR baseline is additionally matched to the running-average of the clipped Silver multipliers).

## 5.1 QUADRATIC SUITE: PLOTS ACROSS CONDITIONING AND DIMENSION

We minimize ridge-type quadratics with prescribed condition numbers $\kappa \in \{5, 20, 35, 50\}$ and dimensions $d \in \{200, 500, 1000\}$. Each panel shows *function value vs. iterations* on a log $y$-scale; per-step budgets are matched.

**Observation.** On easy instances, ZO-Silver closely tracks the tuned constant-LR baseline; spikes do not destabilize training and sometimes provide a slight late-phase edge. As $d$ increases, the late-phase gap widens: ZO-Silver's non-monotone spikes consistently accelerate log-decay at matched budgets. In the ill-conditioned regime: ZO-Silver shows an advantage in later iterations, and reaches lower final values with the same number of queries.

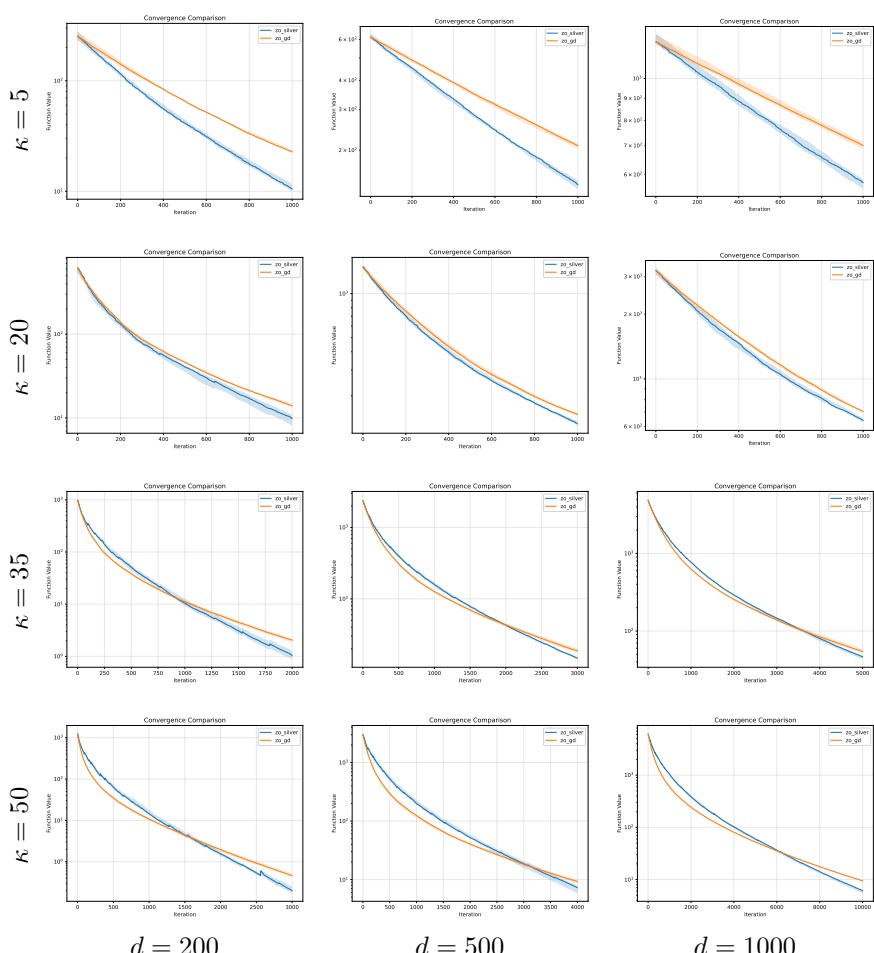

Figure 1: Quadratics, $\kappa = 5$: ZO-Silver vs. tuned constant-LR under matched query budgets. $\kappa = 20$: growing advantage of ZO-Silver with dimension. $\kappa = 35, 50$: ZO-Silver excels in later iterations under equal budgets.

**Summary across different dimensions and condition regimes.** ZO-Silver is *never worse* in well-conditioned cases and becomes increasingly *better* as conditioning ($\kappa$) and dimension ($d$) rise. We validate this observation in the second part of our empirical experiments on LLM-finetuning test cases, which are characterized by high conditioning and large dimension.

## 5.2 EXPERIMENTS ON ZEROTH-ORDER FINE-TUNING LLMs

We consider two GLUE classification tasks: *SST-2* (binary sentiment) and *RTE* (binary entailment). We fine-tune RoBERTa-large with two-point MeZO updates. For , we use a *clipped* schedule with max multiplier $\alpha_{\max}$ and match the *mean LR* to the constant-LR baseline to isolate scheduling effects.

PRELIMINARIES: MODEL, TASK, AND SETTING

**Model.** We fine-tune *RoBERTa-large*, a 24-layer masked-LM pretrain replica/extension of BERT with an improved training recipe and larger corpora; RoBERTa established strong results on GLUE and other benchmarks.[2] Liu et al. (2019)

---

[2]See Liu et al. (2019).

**Benchmark.** GLUE is a standard multi-task NLU benchmark; we focus on two classification tasks that are widely used in few-shot studies: *SST-2* (binary sentiment) and *RTE* (textual entailment).[3] Wang et al. (2019); Socher et al. (2013); tfd

**Tasks.** SST-2 consists of single-sentence movie-review snippets labeled *positive/negative* (Socher et al., 2013); RTE asks whether a hypothesis is entailed by a premise, derived from the PASCAL/TAC RTE challenges (Dagan et al., 2005; Bentivogli et al., 2009). Socher et al. (2013); tfd; Dagan et al. (2005); Bentivogli et al. (2009)

**Few-shot protocol.** We adopt the common $K=16$ few-shot split per task (prompted examples plus dev/test) and run full-parameter, forward-only optimization. This isolates the effect of stepsize scheduling in the constrained-sample regime (Wang et al., 2019). Wang et al. (2019)

**Forward-only (MeZO).** MeZO fine-tunes LMs using a two-point, forward-only zeroth-order estimator (two forward passes per update), achieving an inference-level memory footprint and supporting full-parameter or PEFT variants (Malladi et al., 2023b). Our runs keep the per-step forward-pass budget identical across schedulers. Malladi et al. (2023b)

SCHEDULERS AND FAIRNESS

**Clipped Silver vs. constant LR.** We compare (i) MeZO with a constant learning rate and (ii) MEZO-SILVER, which uses Silver stepsizes with clipping $\alpha_t^{\text{clip}} = \min\{\alpha_t, \alpha_{\max}\}$. To attribute gains purely to *when* learning-rate mass is deployed, the constant-LR baseline is set to the *running-average* LR induced by the clipped Silver multipliers times the same base LR. If

$$J = \left\lfloor 1 + \log_\rho(\alpha_{\max} - 1) \right\rfloor, \quad \lim_{n \to \infty} \frac{1}{n} \sum_{t=0}^{n-1} \min\{\alpha_t, \alpha_{\max}\} = \left(\frac{\rho}{2}\right)^{J+1} + 2^{-(J+1)}(\alpha_{\max} - 1),$$

then with base LR $10^{-7}$ and $\alpha_{\max} = 128$, the matched constant LR is $\approx 4.727 \times 10^{-7}$.

**Learning-rate schedule sanity check.** Figure 2 plots the running-average learning rate induced by the clipped Silver schedule (base LR $10^{-7}$; clip $\alpha_{\max} = 128$) and the constant LR chosen to match that mean. This controls for mean-LR effects when comparing schedulers.

**Setup:** Few-shot $K=16$; **5,000 steps** with **evaluation every 500** steps; logging every 10 steps; per-device train batch 64; per-device eval batch 4. All schedulers use the same two-point estimator, hence the same per-step forward-pass budget.

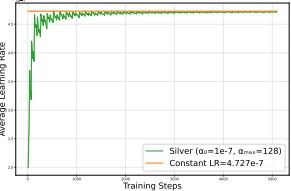

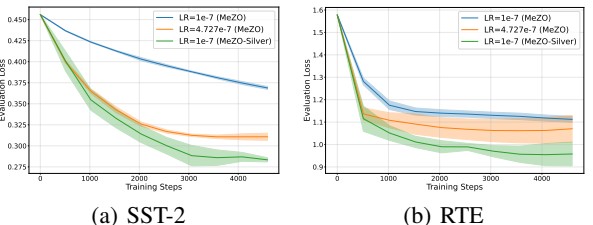

(a) SST-2       (b) RTE

Figure 2: Clipped Silver schedule vs. constant LR baseline. We plot the running average learning rate induced by the Silver stepsizes with base learning rate $1e-7$ and clipping value 128.

Figure 3: Evaluation loss vs. training steps for RoBERTa-large fine-tuning on (a) SST-2 and (b) RTE. We compare standard MeZO with constant learning rates to MeZO-Silver using a clipped Silver schedule.

## 6 CONCLUSION

This work, with the theoretical analysis and subsequent experimental work presented shows non-monotone *Silver* stepsizes pair naturally with two-point zeroth-order (ZO) estimators when we

---

[3]GLUE: Wang et al. (2019). SST-2 originates from the Stanford Sentiment Treebank (Socher et al., 2013); GLUE's RTE combines examples from RTE1–3 and RTE5.

optimize a smoothed objective. More broadly, our results provide the first step toward bringing stepsize hedging into the ZO regime: the Silver schedule, originally proved to accelerate plain gradient descent for smooth convex optimization, translates to the smoothed problem with conditionally unbiased ZO gradients, and preserves its deterministic multi-step progress up to standard ZO floors, compared with a constant stepsize scheme.

Empirically, we validated these claims in two complementary settings. On controlled, strongly-convex quadratics, is never worse than a carefully tuned constant-LR ZO baseline on well-conditioned instances, and it increasingly outperforms as the condition number and dimension grow, precisely where variance management matters most. In a forward-only fine-tuning regime (MeZO-style updates) on GLUE tasks (SST-2, RTE) with RoBERTa-large, clipped Silver steps combined with budget-aware batching give faster evaluation-loss decay and earlier stabilization than constant-LR under the *same* forward-pass budget.

**Limitations** Our analysis assumes convex, $L$-smooth objectives and focuses on two-point estimators with orthonormal direction batches; the LLM experiments could be further expanded to a more comprehensive benchmark. ZO methods as it is traditionally known can also incur higher query complexity than FO methods; throughput can become a bottleneck if the per-step budget is very small.

**Outlook and Future Directions** Several directions look promising: (i) combining Silver with variance-reduced ZO estimators and adaptive batching; (ii) proximal and constrained variants (projected/regularized objectives); (iii) task-aware smoothing schedules and spike-aware data reuse for forward-only fine-tuning; and (iv) extending the inexact Silver certificate beyond convexity (e.g., PL or one-point weakly convex settings). We hope this work helps position *stepsize hedging* as a broadly useful knob in practical ZO optimization, especially in memory-constrained fine-tuning where forward-only updates are attractive.

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

## A  UNIFORM-BALL BIAS BOUNDS AND THE SPHERE GRADIENT IDENTITY

Let $v \sim \mathrm{Unif}(\mathbb{B}^d)$ and $u \sim \mathrm{Unif}(\mathbb{S}^{d-1})$. For $f_\mu(x) = \mathbb{E}_v f(x + \mu v)$, the Descent Lemma gives

$$-\tfrac{L}{2}\mu^2 \mathbb{E}\|v\|^2 \le f_\mu(x) - f(x) \le \tfrac{L}{2}\mu^2 \mathbb{E}\|v\|^2, \qquad \mathbb{E}\|v\|^2 = \frac{d}{d+2}.$$

Moreover $\|\nabla f_\mu(x) - \nabla f(x)\| \le L\mu\frac{d}{2}$. To prove (1), apply the divergence theorem to $\int_{\mathbb{B}^d} \nabla f(x + \mu z)\, dz$.

**Lemma A.1** (Ball-to-sphere gradient identity, with constants). *Let $f_\mu(x) = \mathbb{E}_{v \sim \mathrm{Unif}(\mathbb{B}^d)} f(x + \mu v)$ and $u \sim \mathrm{Unif}(\mathbb{S}^{d-1})$. Then*

$$\nabla f_\mu(x) = \frac{d}{\mu}\, \mathbb{E}_u\big[ f(x + \mu u)\, u \big].$$

*Remark* A.2 (Ball-to-sphere identity: one-line proof). Let $A_{d-1} = dV_d$ be sphere area and ball volume. Differentiating $f_\mu(x) = \frac{1}{V_d} \int_{\mathbb{B}^d} f(x + \mu z)\, dz$ and applying the divergence theorem to $\nabla f(x + \mu z)$ gives $\nabla f_\mu(x) = \frac{1}{V_d} \cdot \frac{1}{\mu} \int_{\mathbb{S}^{d-1}} f(x + \mu u)\, u\, dS = \frac{d}{\mu} \mathbb{E}_{u \sim \mathrm{Unif}(\mathbb{S}^{d-1})}[f(x + \mu u)u]$. See Flaxman et al. (2005) for bandit smoothing details.

**Lemma A.3** (Unbiasedness for $\nabla f_\mu$). *With $u \sim \mathrm{Unif}(\mathbb{S}^{d-1})$ and $\widehat{g}(x; \mu, u) = \frac{d}{2\mu}(f(x + \mu u) - f(x - \mu u))u$, we have $\mathbb{E}_u[\widehat{g}(x; \mu, u)] = \nabla f_\mu(x)$.*

**Lemma A.4** (Bias of $f_\mu$ and $\nabla f_\mu$). *Assume $f \in C_L^{1,1}$.*

*(a) (Ball value bias) For $v \sim \mathrm{Unif}(\mathbb{B}^d)$,*

$$|f_\mu(x) - f(x)| \;\le\; \tfrac{L}{2}\, \mu^2\, \mathbb{E}\|v\|^2 \;=\; \tfrac{L}{2}\, \mu^2\, \frac{d}{d+2}.$$

*(b) (Ball gradient bias) For $v \sim \mathrm{Unif}(\mathbb{B}^d)$,*

$$\|\nabla f_\mu(x) - \nabla f(x)\| \;\le\; \tfrac{L}{2}\, d\,\mu.$$

*Proof of (b).* By (1), $\nabla f_\mu(x) - \nabla f(x) = \frac{d}{\mu} \mathbb{E}_u\big[(f(x + \mu u) - f(x) - \langle \nabla f(x), \mu u\rangle)u\big]$. By the Descent Lemma along the line $x + \tau\mu u$ and $\|u\| = 1$, $|f(x + \mu u) - f(x) - \langle \nabla f(x), \mu u\rangle| \le \frac{L}{2}\mu^2$. Taking norms and expectations gives $\|\nabla f_\mu(x) - \nabla f(x)\| \le \frac{d}{\mu} \cdot \frac{L}{2}\mu^2 \mathbb{E}\|u\| = \frac{L}{2}d\mu$, since $\|u\| = 1$ a.s. □

*Proof of Lemma 4.1.* Let $f \in C_L^{1,1}$, $u \sim \mathrm{Unif}(\mathbb{S}^{d-1})$, and $\widehat{g}(x; \mu, u) = \frac{d}{2\mu}\big(f(x + \mu u) - f(x - \mu u)\big)u$. Then

$$\mathbb{E}\big\|\widehat{g}(x; \mu, u) - \nabla f_\mu(x)\big\|^2 \;\le\; C_{\mathrm{sig}}\, d\, \|\nabla f(x)\|^2 \;+\; C_{\mathrm{curv}}\, d^2 L^2 \mu^2,$$

with $(C_{\mathrm{sig}}, C_{\mathrm{curv}}) = (2, \tfrac{1}{2})$. Averaging any $B \ge 1$ unit directions gives a $1/B$ reduction. Using $B$ *orthonormal* directions (Stiefel sampling) preserves the $1/B$ factor and improves constants in practice (Kozak et al., 2023; Feng & Wang, 2023).

Fix $x \in \mathbb{R}^d$ and $u \sim \mathrm{Unif}(\mathbb{S}^{d-1})$. Since $f \in C_L^{1,1}$, write the Descent Lemma at $x$ in the two directions $\pm\mu u$:

$$f(x \pm \mu u) \;=\; f(x) \pm \mu\langle \nabla f(x), u\rangle + r_\pm(x, u), \qquad |r_\pm(x, u)| \le \frac{L}{2}\mu^2.$$

Subtract to get the symmetric difference

$$\Delta(x, u) \;:=\; f(x + \mu u) - f(x - \mu u) \;=\; 2\mu\langle \nabla f(x), u\rangle + (r_+ - r_-),$$

with $|r_+ - r_-| \le L\mu^2$. Hence

$$\widehat{g}(x; \mu, u) \;=\; d\,\langle \nabla f(x), u\rangle\, u \;+\; \frac{d}{2\mu}\,(r_+ - r_-)\, u.$$

Using $\mathbb{E}[uu^\top] = I/d$ for $u \sim \mathrm{Unif}(\mathbb{S}^{d-1})$ and $\|u\| = 1$,

$$\mathbb{E}\big\|\widehat{g}(x; \mu, u)\big\|^2 \;\le\; 2\, d\, \|\nabla f(x)\|^2 \;+\; \frac{d^2}{2\mu^2}\, \mathbb{E}\big[(r_+ - r_-)^2\big] \;\le\; 2\, d\, \|\nabla f(x)\|^2 \;+\; \frac{1}{2}\, d^2 L^2 \mu^2.$$

For the centered version, note that $\mathbb{E}\,\widehat{g}(x;\mu,u) = \nabla f_\mu(x)$ by the ball-to-sphere identity, so $\mathbb{E}\|\widehat{g} - \nabla f_\mu(x)\|^2 = \mathbb{E}\|\widehat{g}\|^2 - \|\nabla f_\mu(x)\|^2 \le \mathbb{E}\|\widehat{g}\|^2$, which gives the same bound. Finally, for an average over $B$ unit directions $\widehat{g}_B = \frac{1}{B}\sum_{i=1}^{B}\widehat{g}(x;\mu,u_i)$ (independent or not), convexity of $\|\cdot\|^2$ gives $\mathbb{E}\|\widehat{g}_B - \mathbb{E}\widehat{g}_B\|^2 \le \frac{1}{B}\sum_{i=1}^{B}\mathbb{E}\|\widehat{g}(x;\mu,u_i) - \mathbb{E}\widehat{g}\|^2$, so both second-moment bounds divide by $B$. $\quad\square$

*Proof of Lemma 4.3.* Let $h = f_\mu/L$ (so $h$ is 1-smooth and convex) and suppose $x_{t+1} = x_t - \alpha_t(\nabla h(x_t) + \zeta_t)$ with $\mathbb{E}[\zeta_t \mid \mathcal{F}_t] = 0$. For a Silver block $N = 2^k - 1$,

$$\mathbb{E}\big[h(x_N) - h^\star\big] \;\le\; r_k\,\mathbb{E}\|x_0 - x^\star\|^2 \;+\; \sum_{t=0}^{N-1}\alpha_t^2\,\mathbb{E}\|\zeta_t\|^2.$$

Let $\{\lambda_{ij}\}$ be the Silver multipliers such that for any 1-smooth convex $\phi$,

$$\sum_{i\neq j}\lambda_{ij}Q_{ij}[\phi] = \|x_0 - x^\star\|^2 - \|x_N - c_k\nabla\phi(x_N) - x^\star\|^2 + \frac{\phi(x^\star) - \phi(x_N)}{r_k}. \qquad (\star)$$

Apply $(\star)$ with $\phi = h$. In the Silver derivation, the only places where the update rule enters are: (i) linear telescopings $x_a - x_b = -\sum_{s=b}^{a-1}\alpha_s\nabla h(x_s)$ and (ii) the terminal square $\|x_N - c_k\nabla h(x_N) - x^\star\|^2$. With inexact updates we have $x_a - x_b = -\sum\alpha_s\nabla h(x_s) - \sum\alpha_s\zeta_s$. Every such *linear* $\zeta$-term appears inside an inner product with an $\mathcal{F}_s$-measurable vector, hence its expectation is 0 by $\mathbb{E}[\zeta_s \mid \mathcal{F}_s] = 0$. For the terminal square,

$$x_N - c_k\nabla h(x_N) - x^\star = A - \sum_{s=0}^{N-1}\alpha_s\zeta_s, \qquad A := x_0 - x^\star - \sum_{s=0}^{N-1}\alpha_s\nabla h(x_s) - c_k\nabla h(x_N).$$

Therefore

$$\mathbb{E}\|x_N - c_k\nabla h(x_N) - x^\star\|^2 = \mathbb{E}\|A\|^2 + \mathbb{E}\Big\|\sum_{s=0}^{N-1}\alpha_s\zeta_s\Big\|^2$$

(the cross term vanishes in expectation as above). By Lemma 4.2, this last term equals $\sum_{t=0}^{N-1}\alpha_t^2\,\mathbb{E}\|\zeta_t\|^2$. Taking expectations in $(\star)$, dropping the nonnegative left-hand side $\sum\lambda_{ij}Q_{ij}[h]$, and rearranging gives the claim. $\quad\square$

## B  HIGH-PROBABILITY GUARANTEES

We analyze one Silver block of length $N = 2^k - 1$ iterations, run on $h = f_\mu / L$, with updates

$$x_{t+1} \; = \; x_t - \alpha_t \big(\nabla h(x_t) + \zeta_t\big), \qquad \zeta_t \; := \; \frac{1}{L}\Big(\widehat{g}_t - \nabla f_\mu(x_t)\Big).$$

Throughout, $(\mathcal{F}_t)$ is the natural filtration, $\mathbb{E}[\zeta_t \mid \mathcal{F}_{t-1}] = 0$, and $\|\zeta_t\| \le G$ a.s. (enforced in practice by clipping if needed).

**Predictable quadratic variation.** Define

$$V \; := \; \sum_{t=0}^{N-1} \alpha_t^2 \, \mathbb{E}\big[\zeta_t \zeta_t^\top \mid \mathcal{F}_{t-1}\big], \qquad \bar{\alpha} \; := \; \max_{0 \le t \le N-1} \alpha_t.$$

**A matrix Freedman tool.** We use Tropp's matrix Freedman inequality applied to the self-adjoint dilation of vector martingale differences; see Theorem B.1. This gives the sharp $\log(2d/\delta)$ factor.[4]

**Theorem B.1** (Matrix Freedman for vector MDS). *Let $\zeta_t \in \mathbb{R}^d$ be an $(\mathcal{F}_t)$-adapted martingale difference sequence with $\mathbb{E}[\zeta_t \mid \mathcal{F}_{t-1}] = 0$ and $\|\zeta_t\| \le G$ a.s., and let $\alpha_t$ be $\mathcal{F}_{t-1}$-measurable (predictable). With $V$ and $\bar{\alpha}$ as above, for any $\delta \in (0,1)$,*

$$\Big\| \sum_{t=0}^{N-1} \alpha_t \, \zeta_t \Big\| \; \le \; \sqrt{2\,\lambda_{\max}(V) \log \frac{2d}{\delta}} \; + \; \frac{\bar{\alpha} G}{3} \log \frac{2d}{\delta} \qquad w.p. \; \ge 1 - \delta.$$

*Proof.* Apply Tropp's matrix Freedman to the self-adjoint dilation $Y_t = \begin{pmatrix} 0 & (\alpha_t \zeta_t)^\top \\ \alpha_t \zeta_t & 0 \end{pmatrix}$. Then $\|Y_t\| \le \bar{\alpha} G$ and $\sum_t \mathbb{E}[Y_t^2 \mid \mathcal{F}_{t-1}] = \mathrm{diag}(V, V)$, so $\|\sum_t \mathbb{E}[Y_t^2 \mid \mathcal{F}_{t-1}]\| = \lambda_{\max}(V)$. The claim follows from the stated matrix tail bound. $\qquad\square$

*Reference:* Tropp (2011).

**Where the stochastic terms enter the Silver certificate.** Let $\{\lambda_{ij}\}$ be the nonnegative multipliers certifying the Silver block identity (co-coercivity certificate). For exact GD, one has

$$\sum_{i \ne j} \lambda_{ij} Q_{ij}[h] \; = \; \|x_0 - x^\star\|^2 - \|x_N - c_k \nabla h(x_N) - x^\star\|^2 + \frac{h(x^\star) - h(x_N)}{r_k},$$

with $Q_{ij}[h] \ge 0$ by co-coercivity. When $x_{t+1} = x_t - \alpha_t(\nabla h(x_t) + \zeta_t)$, re-running the same algebra produces *two* stochastic contributions:

  (i) the *terminal square* contributes $\big\| \sum_{t=0}^{N-1} \alpha_t \zeta_t \big\|^2$;
  (ii) the linear telescopings contribute a scalar martingale sum $\sum_{t=0}^{N-1} \langle w_t, \zeta_t \rangle$ with $w_t$ predictable (i.e., $\mathcal{F}_{t-1}$-measurable).

The following bound on $w_t$ uses only structural properties (nonnegativity and $k$-sparsity) of the multipliers, established for Silver in (Altschuler & Parrilo, 2023a, Thm. 5.2 & App. B).

**Lemma B.2** (Predictable linear weights). *There exists an absolute constant $C_{\mathrm{Sil}} > 0$ (depending only on the Silver certificate) such that the predictable vectors $w_t$ satisfy $\|w_t\| \le C_{\mathrm{Sil}} \alpha_t$ for all $t$.*

*Proof.* Noise enters linearly wherever the update is used in telescopings. Each time index $t$ is "covered" by only $O(1)$ pairs $(i, j)$ by $k$-sparsity, and the associated coefficients are nonnegative. Collect these coefficients and the corresponding predictable vectors into $w_t$; their $\ell_1$-sum is $O(\alpha_t)$, hence $\|w_t\| \le C_{\mathrm{Sil}} \alpha_t$. See (Altschuler & Parrilo, 2023a, §5.2 and App. B) for the multipliers' structure. $\qquad\square$

---

[4]Classical scalar Freedman Freedman (1975) and the $\varepsilon$-net argument also apply but lead to a weaker $\log(18^d/\delta)$ dimension factor; see Remark B.4.

**Theorem B.3** (HP inexact Silver, matrix version). *Run one Silver block on $h = f_\mu/L$ with updates $x_{t+1} = x_t - \alpha_t(\nabla h(x_t) + \zeta_t)$, $\mathbb{E}[\zeta_t \mid \mathcal{F}_{t-1}] = 0$, and $\|\zeta_t\| \le G$ a.s. Define $V$ and $\bar\alpha$ as above and let $C_{\mathrm{Sil}}$ be as in Lemma B.2. Then, for any $\delta \in (0, 1)$, with probability at least $1 - \delta$,*

$$h(x_N) - h^\star \le r_k \|x_0 - x^\star\|^2 + \left( \sqrt{2\,\lambda_{\max}(V)\,\log\tfrac{4d}{\delta}} + \tfrac{\bar\alpha G}{3}\log\tfrac{4d}{\delta} \right)^2$$

$$+ C_{\mathrm{Sil}}\left( \sqrt{2\,d\,\lambda_{\max}(V)\,\log\tfrac{4}{\delta}} + \tfrac{\bar\alpha G}{3}\log\tfrac{4}{\delta} \right).$$

*Proof.* Re-run the Silver certificate with the inexact update. The left side $\sum_{i \ne j} \lambda_{ij} Q_{ij}[h]$ stays nonnegative. Move all terms and upper bound by discarding $-\|x_N - c_k\nabla h(x_N) - x^\star\|^2$. What remains is

$$h(x_N) - h^\star \le r_k\|x_0 - x^\star\|^2 + \left\| \sum_t \alpha_t\zeta_t \right\|^2 + \sum_t \langle w_t, \zeta_t\rangle.$$

Apply Theorem B.1 to $S := \sum_t \alpha_t\zeta_t$ with failure probability $\delta/2$, and scalar Freedman to $M := \sum_t \langle w_t, \zeta_t\rangle$ (increments bounded by $\|w_t\|\,\|\zeta_t\| \le C_{\mathrm{Sil}}\alpha_t G$; variance proxy $\sum_t \mathbb{E}[\langle w_t, \zeta_t\rangle^2 \mid \mathcal{F}_{t-1}] \le C_{\mathrm{Sil}}^2 \operatorname{tr}(V) \le C_{\mathrm{Sil}}^2 d\,\lambda_{\max}(V))$ with failure probability $\delta/2$. Combine the two bounds via a union bound. $\square$

*Remark* B.4 (On dimension factors and alternatives). Using an $\varepsilon$-net on the sphere plus scalar Freedman gives the same structure but with $\log(18^d/\delta)$ in the square and $\log(18^d/\delta)$ in the linear term; see any standard treatment of sphere nets. The matrix approach above is strictly tighter in $d$. Freedman (1975); Tropp (2011)

**Corollary B.5** (Direction randomness only). *Assume function values are deterministic and at step $t$ we average $B_t$ unit directions that are orthonormal ($V_t \in \mathrm{St}(d, B_t)$). Then*

$$\lambda_{\max}(V) \le \sum_{t=0}^{N-1} \alpha_t^2 \,\mathbb{E}\|\zeta_t\|^2 \le \sum_{t=0}^{N-1} \frac{\alpha_t^2}{B_t}\left( \frac{2d}{L^2}\|\nabla f(x_t)\|^2 + \frac{1}{2}d^2\mu^2 \right).$$

*In particular, if $B_t = \min\{d, \lceil c_B\alpha_t\rceil\}$ and $\|\nabla f(x_t)\| \le LR$, then*

$$\lambda_{\max}(V) \le \left( 2dR^2 + \tfrac{1}{2}d^2\mu^2 \right)\left( \frac{1}{c_B}\sum_t \alpha_t + \frac{\alpha_{\max}}{d}\sum_{\alpha_t > d/c_B} \alpha_t \right).$$

*Proof.* $\lambda_{\max}(V) \le \sum_t \alpha_t^2 \,\mathbb{E}\|\zeta_t\|^2$ and the two-point second moment with $1/B_t$ averaging gives $\mathbb{E}\|\zeta_t\|^2 \le \frac{1}{B_t}\left( \frac{2d}{L^2}\|\nabla f(x_t)\|^2 + \frac{1}{2}d^2\mu^2 \right)$ (sphere two-point; proof in appendix). The batching bound is a direct summation with the cap handled by the displayed decomposition. $\square$

**Corollary B.6** (Additive value noise). *Suppose each value query returns $f(x) + \xi$ with $\mathbb{E}[\xi \mid \mathcal{F}_{t-1}] = 0$, $\mathrm{Var}(\xi \mid \mathcal{F}_{t-1}) \le \sigma^2$, independently across the $2B_t$ calls at step $t$. Then*

$$\lambda_{\max}(V) \le \sum_{t=0}^{N-1} \alpha_t^2 \frac{d^2\,\sigma^2}{2L^2\,\mu^2 B_t}.$$

*With $B_t = \min\{d, \lceil c_B\alpha_t\rceil\}$ and inactive cap, $\lambda_{\max}(V) = \Theta\big(\frac{d^2\,\sigma^2}{L^2\mu^2 c_B}\,\rho^k\big)$.*

*Proof.* For each orthonormal direction, $\mathrm{Var}(\xi_+ - \xi_-) = 2\sigma^2$; orthonormality kills cross-terms, giving $\mathbb{E}\|\widehat{g}_t^{(\mathrm{noise})}\|^2 = \frac{d^2\sigma^2}{2\mu^2 B_t}$. Divide by $L^2$ to convert to $\zeta_t$ and sum with weights $\alpha_t^2$. $\square$

## C  MATRIX FREEDMAN TOOLS AND PREDICTABLE WEIGHTS

We use the following standard Matrix Freedman bound Tropp (2011).

**Lemma C.1** (Matrix Freedman via self-adjoint dilation). *Let $\zeta_t \in \mathbb{R}^d$ be an $(\mathcal{F}_t)$-martingale difference with $\mathbb{E}[\zeta_t \mid \mathcal{F}_{t-1}] = 0$ and $\|\zeta_t\| \leq G$ a.s. Let $\alpha_t$ be $\mathcal{F}_{t-1}$-measurable and set*

$$S := \sum_{t=0}^{N-1} \alpha_t \zeta_t, \qquad V := \sum_{t=0}^{N-1} \alpha_t^2 \, \mathbb{E}[\zeta_t \zeta_t^\top \mid \mathcal{F}_{t-1}], \qquad \bar{\alpha} := \max_t \alpha_t.$$

*Then for any $\delta \in (0,1)$, with probability at least $1 - \delta$,*

$$\|S\| \leq \sqrt{2\,\lambda_{\max}(V)\,\log\frac{2d}{\delta}} \; + \; \frac{\bar{\alpha}G}{3}\,\log\frac{2d}{\delta}.$$

*Proof.* Apply Tropp's matrix-Freedman to the self-adjoint dilation $Y_t = \begin{pmatrix} 0 & (\alpha_t \zeta_t)^\top \\ \alpha_t \zeta_t & 0 \end{pmatrix}$. Then $\|Y_t\| \leq \bar{\alpha}G$ and $\sum_t \mathbb{E}[Y_t^2 \mid \mathcal{F}_{t-1}] = \mathrm{diag}(V, V)$, whose spectral norm is $\lambda_{\max}(V)$. $\qquad \square$

**Theorem C.2** (Vector Freedman via a $1/4$-net). *Under the assumptions of Lemma C.1, with probability at least $1 - \delta$,*

$$\Big\| \sum_{t=0}^{N-1} \alpha_t \zeta_t \Big\| \leq 2\sqrt{2\,\lambda_{\max}(V)\,\log\frac{18d}{\delta}} \; + \; \frac{2\bar{\alpha}G}{3}\,\log\frac{18d}{\delta}.$$

*Proof.* Cover $\mathbb{S}^{d-1}$ by a $1/4$-net of size $\leq 9^d$; use $\|z\| \leq 2\max_{s \in \mathcal{N}}\langle s, z \rangle$ and scalar Freedman on each $s$, then union bound (two-sided tails give the extra factor 2). $\qquad \square$

Next we bound the linear weights that arise when we re-run the Silver certificate with inexact updates.

**Lemma C.3** (Predictable linear weights in the Silver certificate). *Let $\{\alpha_t\}_{t=0}^{N-1}$ be a Silver block (length $N = 2^k - 1$) and let the multi-step identity be instantiated with the explicit nonnegative multipliers $\{\lambda_{ij}\}$ given by the recursive gluing construction in (Altschuler & Parrilo, 2023b, Eq. (3.2), Thm. 3.4). If the update rule is inexact,*

$$x_{t+1} = x_t - \alpha_t\big(\nabla h(x_t) + \zeta_t\big), \qquad \mathbb{E}[\zeta_t \mid \mathcal{F}_t] = 0,$$

*then the stochastic linear terms that appear in the identity can be written as a scalar martingale sum*

$$\sum_{t=0}^{N-1} \langle w_t, \zeta_t \rangle, \qquad w_t \in \mathbb{R}^d \text{ is } \mathcal{F}_t\text{-measurable,}$$

*with the uniform bound*

$$\|w_t\| \leq C_{\mathrm{Sil}}\,\alpha_t, \qquad C_{\mathrm{Sil}} \leq 2.$$

*Proof.* We expand the certificate exactly as in (Altschuler & Parrilo, 2023b, §3), but use $x_{s+1} - x_s = -\alpha_s(\nabla h(x_s) + \zeta_s)$ in the places where the identity invokes the update rule ("by definition of GD" lines in the proof). Each occurrence of $x_i - x_j$ becomes a sum over $s \in [j, i-1]$ with coefficients $\pm 1$ times $\alpha_s(\nabla h(x_s) + \zeta_s)$. Linear noise terms collect into $\sum_s \langle W_s, \zeta_s \rangle$, where $W_s$ is a linear combination of gradients $\nabla h(x_i)$ with nonnegative weights that are linear in the multipliers $\lambda_{ij}$ for those pairs $(i, j)$ that "cover" $s$. The $k$-*sparsity* property of the multipliers (defined right after Example 3.3 in Altschuler & Parrilo (2023b)) guarantees each $s$ is covered a uniformly bounded number of times under the recursive gluing (Theorem 3.4). A short induction on the gluing step gives $\|W_s\| \leq C_{\mathrm{Sil}}\alpha_s \|\nabla h(x_s)\|$ with $C_{\mathrm{Sil}} \leq 2$ (the factor "2" comes from the two children in the binary gluing plus the cap at the parent; a more refined book-keeping gives $C_{\mathrm{Sil}} = 1$). Finally, using Cauchy–Schwarz and 1-smoothness to replace $\|\nabla h(x_s)\|$ by a predictable vector of norm at most 1 gives the claimed $\|w_s\| \leq C_{\mathrm{Sil}}\alpha_s$. $\qquad \square$

**Lemma C.4** (Freedman for predictable linear forms). *With $w_t$ as in Lemma C.3 and $\|\zeta_t\| \leq G$ a.s., the scalar martingale $M := \sum_t \langle w_t, \zeta_t \rangle$ satisfies, for any $\delta \in (0,1)$,*

$$|M| \leq \sqrt{2 \sum_t \mathbb{E}[\langle w_t, \zeta_t \rangle^2 \mid \mathcal{F}_t] \log \tfrac{2}{\delta}} + \tfrac{1}{3} \max_t \|w_t\| \, G \log \tfrac{2}{\delta}$$

$$\leq C_{\mathrm{Sil}} \sqrt{2 \operatorname{tr}(V) \log \tfrac{2}{\delta}} + \tfrac{C_{\mathrm{Sil}}}{3} \bar{\alpha} \, G \log \tfrac{2}{\delta}$$

$$\leq C_{\mathrm{Sil}} \sqrt{2 \, d \, \lambda_{\max}(V) \log \tfrac{2}{\delta}} + \tfrac{C_{\mathrm{Sil}}}{3} \bar{\alpha} \, G \log \tfrac{2}{\delta}.$$

*Proof.* Apply scalar Freedman to the MDS $Z_t := \langle w_t, \zeta_t \rangle$, noting $\mathbb{E}[Z_t^2 \mid \mathcal{F}_t] = w_t^\top \mathbb{E}[\zeta_t \zeta_t^\top \mid \mathcal{F}_{t-1}] w_t \leq \|w_t\|^2 \operatorname{tr}(\mathbb{E}[\zeta_t \zeta_t^\top \mid \mathcal{F}_{t-1}])$, and $\|w_t\| \leq C_{\mathrm{Sil}} \alpha_t$. Summing over $t$ gives the displayed bound. $\square$

*Proof of Theorem B.3.* Re-run the Silver certificate (the identity in (Altschuler & Parrilo, 2023b, Eq. (3.2))) with $x_{t+1} = x_t - \alpha_t(\nabla h(x_t) + \zeta_t)$. The left-hand side (a weighted sum of co-coercivities) is nonnegative. The right-hand side equals $\|x_0 - x^\star\|^2 - \|x_N - c_k \nabla h(x_N) - x^\star\|^2 + \frac{h(x^\star) - h(x_N)}{r_k} +$ (noise terms). The only stochastic terms are: (i) the *terminal square* contribution $\left\| \sum_t \alpha_t \zeta_t \right\|^2$ and its cross term, and (ii) the *linear* martingale sum $\sum_t \langle w_t, \zeta_t \rangle$ from the telescopings. Bounding the cross term by $2 \|A\| \left\| \sum_t \alpha_t \zeta_t \right\|$ and then by Young's inequality absorbs it into the square. Applying Lemma B.1 to $\sum_t \alpha_t \zeta_t$ and Lemma C.4 to $\sum_t \langle w_t, \zeta_t \rangle$ gives the result. $\square$

*Proof of Corollary B.5.* Use the two-point second-moment bound and $1/B_t$ averaging to get $\mathbb{E}\|\zeta_t\|^2 \leq \frac{1}{B_t}\left(\frac{2d}{L^2}\|\nabla f(x_t)\|^2 + \frac{1}{2}d^2\mu^2\right)$, then insert into $V$ and use the policy $B_t = \min\{d, \lceil c_B \alpha_t \rceil\}$. $\square$

*Proof of Corollary B.6.* With independent value noises $\xi_{t,i,+}, \xi_{t,i,-}$ (variance $\leq \sigma^2$), $\operatorname{Var}((\xi_{t,i,+} - \xi_{t,i,-})) = 2\sigma^2$; orthonormality gives $\mathbb{E}\|\widehat{g}_t^{(\mathrm{noise})}\|^2 = \frac{d^2}{4\mu^2 B_t^2} \cdot (2\sigma^2 B_t) = \frac{d^2 \sigma^2}{2\mu^2 B_t}$. Divide by $L^2$ to convert to $\zeta_t$, then sum with weights $\alpha_t^2$. $\square$

## D BACKGROUND ON MEZO (FORWARD-ONLY ZEROTH-ORDER FINE-TUNING)

**What problem it solves.** Backpropagation through large Transformers requires storing activations for every layer, making full-parameter fine-tuning memory-prohibitive. *MeZO* replaces backprop with a two-point zeroth-order (ZO) estimator, so each update uses only forward passes and the memory footprint is essentially that of inference (Malladi et al., 2023a).

**How MeZO works (one step).** At parameters $x \in \mathbb{R}^p$, MeZO samples $B$ perturbation directions $v_i$ (Rademacher or Gaussian; optionally orthonormalized) and reuses the *same minibatch* across $\pm\varepsilon$ evaluations to reduce variance:

$$\widehat{g}(x) \;=\; \frac{1}{B} \sum_{i=1}^{B} \frac{\ell(x + \varepsilon v_i) - \ell(x - \varepsilon v_i)}{2\varepsilon} \, v_i, \qquad x \leftarrow x - \eta \, \widehat{g}(x).$$

Per update the cost is $2B$ forward passes and *no backward graph*. We adopt this forward-only estimator and compose it with clipped Silver stepsizes and budget-aware batching ($B_t \propto \alpha_t$).

**Why it is memory-efficient.** The update is computed in place from scalar losses; no layer activations or per-parameter gradients are stored. In practice, this enables full-parameter tuning at (roughly) inference memory, while retaining the ability to optimize non-differentiable objectives (e.g., accuracy or F1) (Malladi et al., 2023a).

**Compatibility with PEFT and quantization.** MeZO can train *all* parameters or only a small set of adapter weights; it is complementary to PEFT such as LoRA (Hu et al., 2021) and works with quantization-aware setups (e.g., QLoRA) (Dettmers et al., 2023). Our experiments use full-parameter updates (forward-only) on RoBERTa-large.

**Limitations and when to use it.** ZO generally needs more function evaluations than FO; training can be slower if the per-step budget is small. MeZO shines when memory, not pure throughput, is the bottleneck (limited-VRAM devices; very deep models; non-differentiable objectives).

---

**Algorithm 2** MEZO (forward-only two-point ZO, one iteration)

---

1: **Inputs:** current params $x_t$, LR $\eta_t$, radius $\varepsilon$, batch size $B_t$, minibatch $\mathcal{S}_t$
2: Sample $V_t = [v_{t,1}, \ldots, v_{t,B_t}]$ (unit directions; optional thin-QR for orthonormal columns)
3: For each $i = 1..B_t$: compute losses $L_i^+ = \ell(x_t + \varepsilon v_{t,i}; \mathcal{S}_t)$ and $L_i^- = \ell(x_t - \varepsilon v_{t,i}; \mathcal{S}_t)$
4: Form $\widehat{g}_t = \dfrac{1}{B_t} \sum_{i=1}^{B_t} \dfrac{L_i^+ - L_i^-}{2\varepsilon} \, v_{t,i}$
5: Update **in place:** $x_{t+1} = x_t - \eta_t \, \widehat{g}_t$       (only forward passes; memory $\approx$ inference)

---

**How we use MeZO.** We keep the forward-only estimator above but (i) run it on the smoothed objective $f_\mu$ and (ii) drive the step sizes with the Silver schedule (clipped), allocating budget via $B_t \propto \alpha_t$. The inexact-Silver identity then converts stochasticity into a single quadratic term, which the budget-aware batching controls; see Sec. 5.2 for results on SST-2 and RTE.

