# OpenReview forum: "Silver Stepsize for Faster Zeroth-Order Optimization"
_ICLR.cc/2026/Conference — ICLR 2026 Conference Withdrawn Submission_

### Official Review · Reviewer_27zL · 2025-10-15

**Soundness:** 1
**Presentation:** 2
**Contribution:** 1
**Rating:** 2
**Confidence:** 4

**Summary:**

This paper proposes a new stepsize scheduling approach for solving the smooth convex optimization problem, called Silver stepsize. This method is claimed to be faster than standard approaches. Multiple empirical experiments are taken to validate the result.

**Strengths:**

The experimental results seem to have significant improvements; that is, the proposed method indeed improves the given baseline. Both synthetic and practical experiments are considered.

**Weaknesses:**

This paper is apparently incomplete. It shouldn't have been submitted.

Minor: It has unlabeled reference on page 2: ".Section??developstheinexact-gradient".

1. The author didn't list any assumptions and didn't explain notations and the problem setup.
2. The author didn't include any proof for its lemmas and theorems. No additional supplymentary materials or appendices provided. Only proof sketch.
3. This paper only considers the smooth convex case, which has been well explored.
4. The Silver stepsize is not new. The author doesn't justify why it is suitable for gradient-free optimization.

**Questions:**

The current submission is not complete, as no formal proof provided for the main theoretical results. I hope the author include these materials in another submission. For the current one, I suggest rejecting this paper.

---

### Official Review · Reviewer_oiV7 · 2025-10-20

**Soundness:** 2
**Presentation:** 1
**Contribution:** 1
**Rating:** 0
**Confidence:** 3

**Summary:**

This paper proposes a new stepsize scheduling strategy, termed the Silver Stepsize, for accelerating zeroth-order (ZO) optimization. The idea is inspired by a “Silver identity” and involves running blocks of updates called “Silver blocks,” with the goal of improving convergence under limited query budgets. The authors claim that their method achieves better performance than existing ZO optimizers such as MeZO.

**Strengths:**

* Interesting attempt to improve ZO learning rates: The idea of constructing structured step-size schedules or blocks to reduce variance in ZO methods could be valuable if properly formalized and justified.
* Potential theoretical contribution: The paper hints at a nontrivial theoretical structure (“Silver identity” and block recursion), which, if made rigorous, could yield new insights for ZO optimization.

**Weaknesses:**

$\textbf{Logical gaps and unclear derivations}$

* line-163: The derivation of $r_{k}$ is not explained. Simply stating “with explicit” provides no help to the reader or reviewer. It appears that crucial steps are omitted, leaving the correctness unverifiable.

* line-171: The phrase “by the Descent Lemma” is used without specifying which lemma. If it refers to the classical descent lemma for smooth functions, please state it explicitly

Overall, the logic jumps too much, and  key steps are skipped or implied without justification.

---

$\textbf{Not self-explanatory terminology}$
* The manuscript assumes readers already understand concepts such as stepsize hedging (Silver), Silver identity, and Silver block, yet none of these are properly defined or cited.
* For example, Line 145: “we run a Silver block of length $N =2^{k} - 1$", this is not interpretable without knowing what a “Silver block” is or how it is constructed.
* The text thus becomes inaccessible to readers who have not read the previous work. The lack of self-contained definitions makes the paper confusing and non-reviewable.

---

$\textbf{Inconsistent and unclear notation}$
* Line 131 uses $\mu v$ but later Line 135 and beyond switch to $\mu u$ without explanation.
* The notation $\text{St}(d, B_{t})$ is undefined. If this represents a Stiefel manifold (orthonormal columns of $V_{t}$, this must be explicitly stated.
* These inconsistencies make the technical arguments hard to follow and the results unverifiable.

$\textbf{Experimental evaluation}$
* $\textbf{Unfair comparison:}$ \
The proposed method and MeZO differ substantially in computational cost per training step. Reporting results per iteration rather than per query makes the comparison misleading. For ZO methods, query complexity (number of function evaluations) is the standard metric.
* $\textbf{Limited baselines and datasets:}$ \
The paper only reports results on small, toy-like problems. Many recent ZO fine-tuning methods show significant improvements over MeZO but are not included for comparison.
* $\textbf{Hyperparameter fairness:}$ \
For MeZO, the learning rate is taken directly from the proposed method’s schedule without tuning. A proper grid search for the best learning rate should have been conducted.

**Questions:**

$\textbf{Algorithmic clarity}$

* Line-4 of Algorithm 1: Sampling mutually orthonormal columns $V_{t}$ can be computationally expensive, comparable to or exceeding a forward pass for deep models. The paper should discuss this cost explicitly.
* Line-3 of Algorithm 1: How is $L$ estimated or set, and how is $\alpha_{t}$ chosen? These hyperparameters are not explained.
* The number of additional hyperparameters introduced compared to MeZO should also be clearly listed.

---

$\textbf{Theoretical concerns}$

* $\textbf{Theorem 4.5 and Lemma 4.9}$: Both claim upper bounds, but can the upper bound potentially be arbitrarily small?  If so, please justify.
* $\textbf{Lemma 4.9}$: If I understand it correctly, the upper bound appears $d$-dependent.

---

### Official Review · Reviewer_XZvk · 2025-10-31

**Soundness:** 2
**Presentation:** 2
**Contribution:** 3
**Rating:** 4
**Confidence:** 2

**Summary:**

This paper proposes the use of the Silver stepsize schedule, an explicit fractal non-monotone stepsize sequence, to accelerate zeroth-order (gradient-free) optimization of smooth convex functions. The authors extend the multi-step Lyapunov analysis underlying the Silver schedule to settings using unbiased two-point zeroth-order estimators on smoothed objectives, specifically employing an "orthogonal-on-spikes" batching policy to optimally control estimator variance under fixed query budgets. Theoretical results show that the Silver identity remains valid with a quadratic variance cost, and practical experiments—including fine-tuning of large language models (RoBERTa-large) using MeZO-style forward-pass-only schemes—demonstrate improved convergence and stability relative to constant stepsize baselines.

**Strengths:**

1. The paper rigorously adapts the Silver stepsize analysis to the stochastic, zeroth-order regime and transparently derives error bounds using the Silver identity, variance-optimal batching, and martingale concentration.

2. The technical arguments are compact and accessible, with careful attention paid to unbiasedness properties (Lemma 4.2, Lemma A.2) and precise variance analysis (Appendix B). The paper includes direct proofs or proof sketches for all major claims, including second-moment bounds and bias calculations.

3. The paper provides convincing empirical results on synthetic convex problems and on forward-only fine-tuning of LLMs, demonstrating more stable convergence and lower validation loss with the Silver schedule compared to constant stepsizes.

**Weaknesses:**

1. Limited empirical benchmarking breadth and depth.
While the reported results demonstrate robust improvements on both synthetic convex problems and RoBERTa-large fine-tuning, the empirical evaluation remains relatively narrow in scope. The LLM experiments are restricted to 16-shot fine-tuning on only two GLUE tasks (SST-2 and RTE) and a single model architecture. Moreover, Figure 3 reports only the fine-tuning loss curves, which are insufficient to substantiate the claimed performance gains. Since MeZO serves as one of the baselines for this work, the authors should at least present direct comparisons on downstream task metrics—as done in the original MeZO paper—rather than solely showing loss trajectories. Additionally, it would strengthen the paper to include experiments on other model scales or architectures such as GPT-2, LLaMA, or the OPT family used in MeZO.

2. Incompleteness of variance-reduction and acceleration baselines.
In the experiments on synthetic convex problems, the authors compare primarily against standard ZO-GD, which is insufficient to convincingly demonstrate the effectiveness of the proposed method. Although I am not deeply familiar with the specific two-point ZO estimation paradigm, it seems that the paper should include comparisons with more advanced accelerated or variance-reduced zeroth-order optimizers—for example, algorithms that incorporate momentum, adaptive smoothing, or gradient-tracking techniques. Such baselines would provide a stronger empirical validation of the claimed acceleration and variance-control benefits.

3. Minor errors. The reference at line 88 (“section.??”) did not compile correctly.

4. Missing references.
ZO optimization has recently attracted increasing attention for its suitability in memory-constrained large language model fine-tuning, due to its gradient-free nature. Although most recent ZO methods adopt multi-point estimators and are not directly comparable to the approach proposed in this paper, the authors should still pay attention to recent developments (e.g., [1, 2, 3]) and refer to their experimental settings to strengthen and contextualize the empirical section.

[1]. Wang X, Qin X, Yang X, et al. Relizo: Sample reusable linear interpolation-based zeroth-order optimization[J]. Advances in Neural Information Processing Systems, 2024, 37: 15070-15096.

[2]. Shu Y, Zhang Q, He K, et al. Refining adaptive zeroth-order optimization at ease[J]. arXiv preprint arXiv:2502.01014, 2025.

[3] Dang S, Guo Y, Zhao Y, et al. FZOO: Fast Zeroth-Order Optimizer for Fine-Tuning Large Language Models towards Adam-Scale Speed[J]. arXiv preprint arXiv:2506.09034, 2025.

**Questions:**

1. The method is evaluated with two-point ZO estimators. Would the Silver schedule and the “orthogonal-on-spikes” batching strategy generalize to multi-point estimators? If not, could the authors elaborate on the challenges?

2. Can the authors provide or cite additional results on standard accuracy metrics (e.g., classification accuracy, F1) for the LLM experiments in Section 5.2? In addition, it would be valuable to include evaluations on a broader set of benchmarks and across different model architectures to better assess the generality of the proposed method.

3. Could the authors provide a more detailed ablation or empirical study on the influence of block length, smoothing radius schedule ($\mu$), and step size clipping on both convergence and final solution quality?

---

### Official Review · Reviewer_zM9e · 2025-11-01

**Soundness:** 3
**Presentation:** 3
**Contribution:** 2
**Rating:** 4
**Confidence:** 3

**Summary:**

This paper proposes ZO-Silver, a zeroth-order (ZO) optimization method that combines the Silver stepsize schedule with two-point gradient estimators on a smoothed objective. The authors adapt the multi-step Lyapunov analysis of the Silver schedule to the ZO setting and introduce an "orthogonal-on-spikes" batching strategy that allocates query budgets proportionally to step sizes. Theoretically, they show that the method retains the accelerated convergence rate of Silver stepsizes up to a variance term, which is optimally controlled via their batching policy. Empirically, they validate the approach on synthetic problems and in a MeZO-style fine-tuning setting with large language models.

**Strengths:**

++ The work combines stepsize scheduling and zeroth-order optimization, and offers a practical, memory-efficient method for fine-tuning large models where backpropagation is infeasible. The high-probability analysis via Freedman’s inequality is also a nice addition.

**Weaknesses:**

-- The novelty is incremental since the core idea, applying Silver stepsizes to ZO, is a straightforward combination of two existing techniques. While the batching policy is new, the overall conceptual leap is modest.

-- The experiments are relatively narrow. Synthetic tests are limited to ridge regression and logistic regression, and the LLM fine-tuning is only shown on RoBERTa-large for two GLUE tasks. I suggest the authors conduct experiments on more optimization tasks.

-- The baselines are limited. I suggest the authors compare with more ZO optimizers.

**Questions:**

Please see the weaknesses.

---

### Author Response · Authors · 2025-11-28

**We thank the reviewers and the AC for the detailed and thoughtful feedback on our submission _“Silver Stepsize for Faster Zeroth-Order Optimization”_.**
We carefully went through all comments and we agree with the overall assessment that, in its current form, the paper is not yet ready for acceptance.

Motivated by the reviews, we have prepared an updated version of the manuscript (“v2”) that substantially revises the presentation and clarifies the scope of our contributions. In particular, the new version:

- Streamlines and shortens the exposition (especially the introduction and related work) to better highlight the core technical ideas.
- Clarifies the precise optimization setting: convex, L-smooth objectives, uniform-ball smoothing, and two-point estimators with orthogonal directions sampled on the Stiefel manifold.
- Makes more explicit how the inexact-gradient Silver identity is adapted to the zeroth-order setting, and specifies the role of conditional unbiasedness and the quadratic-variance term.
- Tightens and reorganizes the variance analysis, including the use of orthogonal-on-spikes batching $B_t \propto \alpha_t$ under a fixed query budget, and separates what is *provable* from what is *heuristic*.
- Cleans up several technical points in the appendices (smoothing bias bounds, two-point second-moment bounds, and high-probability guarantees) and improves notation throughout.
- Revises the empirical section, focusing on two complementary settings: well-controlled quadratics under varying condition number and dimension, and MeZO-style forward-only fine-tuning on GLUE tasks (SST-2, RTE) with RoBERTa-large, always under matched query budgets.

We acknowledge that in its original form, the submission did not make these aspects sufficiently clear and that some of the broader empirical claims were not supported as convincingly as they should have been. The revised version aims to address these concerns by:

- Being more precise about the claimed benefits of Silver stepsizes in the zeroth-order regime.
- Emphasize the *conceptual* message: that the Silver multi-step Lyapunov identity is robust to conditionally unbiased two-point ZO estimators on a smoothed objective, and that the resulting variance term can be controlled in a budget-aware way via orthogonal-on-spikes batching.
- Present the LLM experiments as a proof-of-concept rather than as a comprehensive empirical benchmark.

Given the reviews and the overall ranking for this cycle, we do not intend to further contest the decision. Instead, we would like to use this comment simply to:

1. Acknowledge the helpful feedback from the reviewers, and
2. Propose a revised “v2” of the paper that incorporates the above changes, for anyone interested in this line of work, while recognizing that the extent of these revisions goes beyond what is reasonable to ask reviewers to re-evaluate at this stage.

Accordingly, we would like to **formally withdraw this submission**. We plan to further polish and consolidate the results (especially the empirical section and connections to broader ZO/PEFT practice) before resubmitting to a future venue.

We thank the reviewers again for their careful evaluations and for their helpful feedback to improve the clarity and scope of this work.

*The Authors*

---

### Note · Authors · 2025-12-03

I have read and agree with the venue's withdrawal policy on behalf of myself and my co-authors.